# The positioning mechanics of microtubule asters in *Drosophila* embryo explants

**Jorge de-Carvalho[1], Sham Tlili[2†], Timothy E Saunders[2,3,4]\*, Ivo A Telley[1]\***

[1]Instituto Gulbenkian de Ciência, Fundação Calouste Gulbenkian, Oeiras, Portugal; [2]Mechanobiology Institute and Department of Biological Sciences, National University of Singapore, Singapore, Singapore; [3]Institute of Molecular and Cellular Biology, A\*Star, Proteos, Singapore, Singapore; [4]Centre for Mechanochemical Cell Biology, Warwick Medical School, University of Warwick, Warwick, United Kingdom

**\*For correspondence:**
timothy.saunders@warwick.ac.uk (TES);
itelley@igc.gulbenkian.pt (IAT)

**Present address:** [†]Aix-Marseille Université | AMU · Institut de Biologie du Développement de Marseille-Luminy (UMR 7288 IBDML), Marseille, France

**Competing interest:** The authors declare that no competing interests exist.

**Abstract** Microtubule asters are essential in localizing the action of microtubules in processes including mitosis and organelle positioning. In large cells, such as the one-cell sea urchin embryo, aster dynamics are dominated by hydrodynamic pulling forces. However, in systems with more densely positioned nuclei such as the early *Drosophila* embryo, which packs around 6000 nuclei within the syncytium in a crystalline-like order, it is unclear what processes dominate aster dynamics. Here, we take advantage of a cell cycle regulation *Drosophila* mutant to generate embryos with multiple asters, independent from nuclei. We use an ex vivo assay to further simplify this biological system to explore the forces generated by and between asters. Through live imaging, drug and optical perturbations, and theoretical modeling, we demonstrate that these asters likely generate an effective pushing force over short distances.

## eLife assessment

This manuscript utilizes a *Drosophila* explant system and modeling to provide **important** insights into the mechanism of microtubule aster positioning. Although the intellectual framework of aster positioning has been worked out by the same authors in their previous work, this study provides additional **solid** evidence to solidify their model.

## Introduction

Eukaryotes assemble a cytoskeletal structure of microtubules (MTs) called an 'aster', which is involved in critical cell functions including intracellular positioning and organelle transport. In metazoan cells, the aster acquires a radial shape in which MTs are focused by the centrosome and emanate toward the cell periphery (*Wilson, 1986*). Microtubules grow at the centrosome and at microtubule-based nucleation sites (*Ishihara et al., 2016*). The centrosome is also found at the two focus points of the mitotic spindle – the poles – linking the aster structurally and mechanically to the spindle, thus lending it a decisive role during cell division (*Hinchcliffe and Sluder, 2001*; *Hoffmann, 2021*). Positioning of the spindle by astral microtubules contacting the cell cortex determines the cell wall cleavage location in sand dollar and *Xenopus* eggs (*Field et al., 2015*; *Mitchison et al., 2012*; *Rappaport, 1969*; *Rappaport, 1961*). During fertilization, the maternal and paternal genomes are united by the assembly and migration of the sperm aster, which facilitates transport of the female toward the male pronucleus in sand dollar and *Drosophila* eggs (*Chambers, 1939*; *Hamaguchi and Hiramoto, 1986*; *Riparbelli et al., 2000*). Where the sperm aster positions inside the egg will roughly define the geometry of the first embryonic cleavage (*Albertson, 1984*; *Hamaguchi and Hiramoto, 1980*; *Hirano and Ishikawa, 1979*). Interestingly, in large egg cells, for example, from

*Xenopus*, the aster migration can also occur without microtubules contacting the cell periphery (*Wühr et al., 2010*). Uninuclear cells contain one or, at most, two asters depending on the cell cycle stage.

The mechanics of aster movement has been a matter of long discourse, so far without any consensus on a unified biophysical model (*Deshpande and Telley, 2021b*). Disagreement remains likely because the experimental insight stems from different model organisms and cells which, by nature, have evolved divergent mechanisms of aster positioning. One fundamental question, whether asters position themselves by pulling on or by pushing at the rigid cell periphery, remains extensively debated (*Garzon-Coral et al., 2016*; *Grill and Hyman, 2005*; *Meaders et al., 2020*; *Sulerud et al., 2020*). Recently, an adaptation of the pulling model for cortex contact-free asters in large cells has gained new momentum (*Hamaguchi and Hiramoto, 1980*; *De Simone et al., 2018*; *Tanimoto et al., 2018*; *Xie et al., 2022*). In this model, the net force moving the aster originates from a balance of viscous drag forces of moving organelles along astral microtubules. Overall geometric asymmetry of the aster – caused by regional differences in microtubule lengths – is a key point of this model (*Tanimoto et al., 2016*).

An interesting yet understudied case of higher complexity is the positioning of multiple nuclei and their associated asters in multinucleated cells (coenocytes). In the syncytial embryo of insects, such as the fruit fly, the genome is rapidly proliferated and distributed without cell cleavage, leading to a single large cell with hundreds of nuclei (*Donoughe and Extavour, 2016*; *Foe and Alberts, 1983*; *Sommer and Tautz, 1991*). The positioning of these syncytial nuclei depends on the centrosome-nucleated microtubule aster (*de-Carvalho et al., 2022*; *Megraw et al., 1999*; *Telley et al., 2012*) and microtubule-associated crosslinking proteins (*Deshpande et al., 2021a*), because perturbation of these components causes aberrant nuclear movements, irregular nuclear distribution, and spindle aggregation. Therefore, nuclear positioning must be mechanistically linked to the separation of neighboring asters (i.e., not being part of the same mitotic spindle). Because these nuclei are not isolated by a cell wall, unlike in uninuclear cells, their positioning occurs relative to the cell cortex *and* each (direct) neighbor. Cell cortex pushing or pulling – whichever applies – is now combined with aster–aster interactions, adding considerable complexity to the mechanical system. Ultimately, a net force is required such that each aster is evenly and stably separated from its neighbors, of which there are on average six (*Kanesaki et al., 2011*). Recent work from us has suggested that short-ranged aster repulsion positions nuclei in a regular order within the syncytial embryo (*de-Carvalho et al., 2022*). However, experimental testing of these forces is still missing.

Here, by exploiting embryonic explants (*Telley et al., 2013*), which reduces complexity, and a cell cycle regulation mutant (*Freeman et al., 1986*; *Lee et al., 2003*) to uncouple microtubule organization from nuclear division, we have studied the dynamics of aster–aster interaction bottom-up. In this system, we uncover the physical principles of separation for simple aster arrays, reveal the positional autonomy of single asters, and derive the microtubule-associated mechanical separation potential. This work builds on our recent study (*de-Carvalho et al., 2022*) and reveals the underlying local mechanics between the structures that lend order to a rapidly proliferating embryonic system.

## Results

Asters are radial microtubule structures nucleated by the centrosome, which acts as microtubule organizing center (MTOC) as part of the mitotic spindle pole. In the early *Drosophila* embryo, the MTOC is physically connected to the nucleus throughout the cell cycle. We wanted to uncouple nucleus- and aster-associated forces and study isolated asters. To this end, we took advantage of *giant nuclei* (*gnu*) mutant *Drosophila* embryos, in which DNA endoreplication occurs without mitosis as the cell cycle is arrested in interphase. This results in the embryo having only one or just a few polyploid nuclei. Yet, centrosome maturation and duplication continue in *gnu* embryos (*Freeman et al., 1986*; *Freeman and Glover, 1987*; *Lee et al., 2003*; *Figure 1A*). We produced embryo explants from *gnu* mutant embryos (*Telley et al., 2013*; *Figure 1B*) and studied the positioning properties of individual or a small number of microtubule asters in the absence of nuclei. These explants exhibit a high aspect ratio with circular planar shape and <10 μm peak height. Thus, for the purpose of kinematic analysis we treated them as quasi-2D spaces.

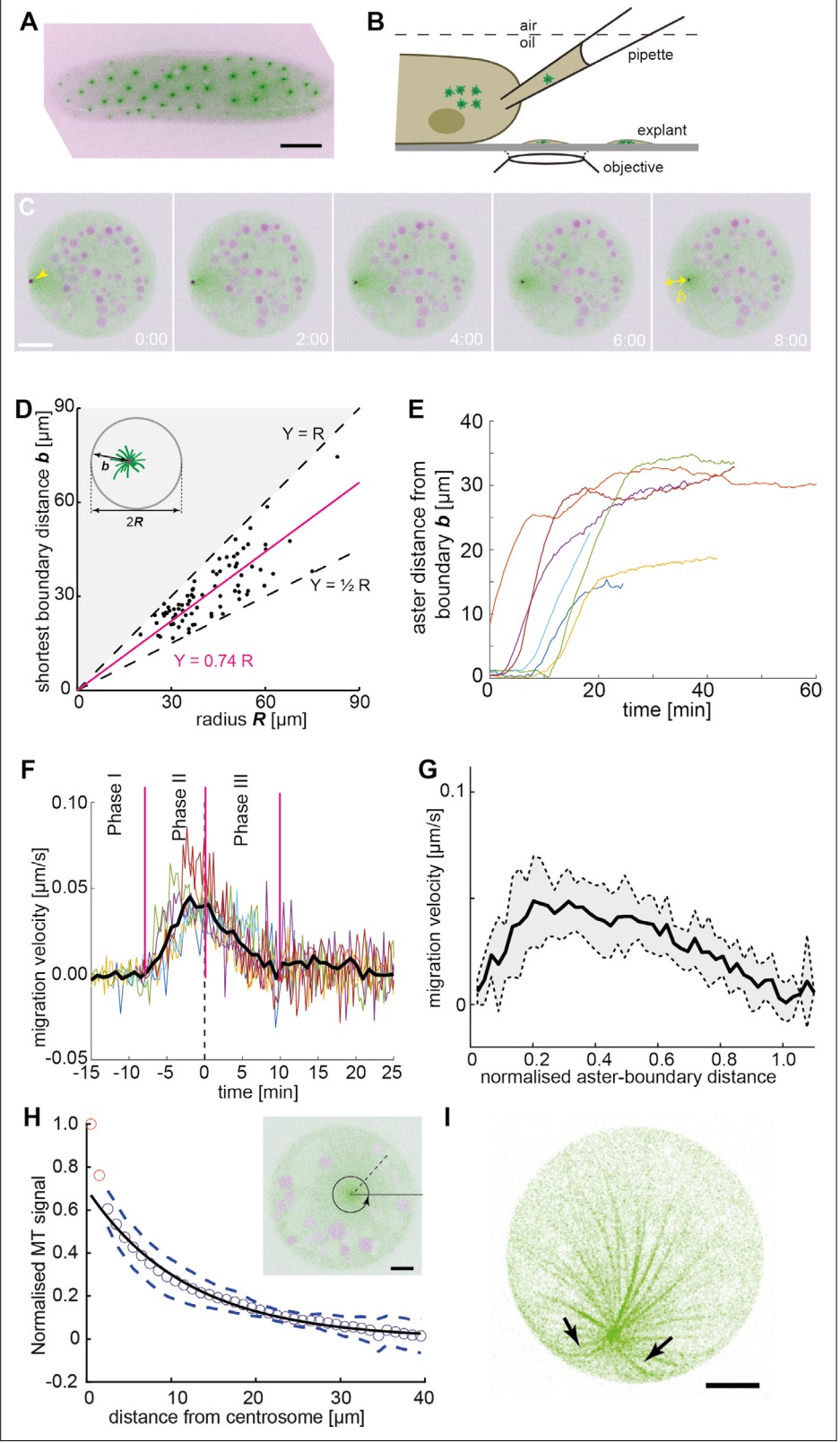

**Figure 1.** Embryo explant assay enables the kinematic study of individual asters. (**A**) Maximum intensity Z-projection of a *gnu* mutant embryo expressing RFP::β-Tubulin (green) and Spd2::GFP (black dots). Scale bar, 50 μm. (**B**) Schematic of cytosol extraction from a *gnu* mutant *Drosophila* embryo and ex vivo explant formation (*de-Carvalho et al., 2018*). (**C**) Maximum intensity Z-projections of a single aster (arrowhead) moving away

*Figure 1 continued on next page*

*Figure 1 continued*

from the boundary of an explant produced from *gnu* mutant embryos expressing RFP::β-Tubulin (green) and Spd2::GFP (black dot). Yolk spheres are visible in magenta due to their auto-fluorescence. In the last frame, the shortest distance *b* from the explant boundary is marked with a yellow double arrow. Scale bar, 20 μm. (**D**) Scatter plot of shortest distance *b* to explant boundary as a function of the radius *R* in explants containing one aster (*n* = 54). The magenta line represents a linear regression. Black dashed lines represent half and full radius distance (the geometric constraint in the system). (**E**) Trajectories of aster distance to the explant boundary from independent experiments. (**F**) Migration velocity as a function of time, where *t* = 0 is defined as the time when the aster lies midway between the explant edge and the final position of the aster. Solid line represents average over all measurements (*n* = 7). (**G**) Average migration velocity of single asters away from the explant boundary (*n* = 7). Distance normalized by the final, steady-state distance for each aster. (**H**) Normalized intensity of astral microtubules as schematically outlines in the inset. The black line is a mono-exponential fit to the data excluding the first two data points (red), representing the centrosome, and the dashed lines mark ±1 standard deviation (SD). The decay length is 11.8 ± 0.5 μm (mean ± standard error of the mean [SEM]), and the intensity drops to background level at ~40 μm. Inset: Single Z-plane image of an explant from a *gnu* mutant embryo expressing RFP::β-Tubulin (green) and Spd2::GFP (black dot), containing a single aster. The dashed black line and the circular arrow represent the radial maximum intensity projection of the microtubule signal from the centrosome toward the periphery aiming at measuring aster size. Scale bar, 10 μm. (**I**) Maximum intensity Z-projection of a 3D image stack of a small explant containing one aster that exemplifies microtubule buckling and splay near the explant boundary (arrows). Scale bar, 5 μm.

The online version of this article includes the following figure supplement(s) for figure 1:

**Figure supplement 1.** Average microtubule signal intensity (black line, inferred from RFP::β-Tubulin signal) along the shortest distance from the centrosome to explant boundary, normalized by the maximum intensity within each experiment.

## Single aster positioning is consistent with radially symmetric forces

Initially, we focused on explants containing a single aster to gain a deeper understanding of the aster–boundary interactions. Here, we note that the explant boundary likely does not have all the properties of a cell cortex, and likely acts more like a (semi) rigid barrier. In these experiments, a single aster consistently moved away from the explant boundary and eventually adopted a steady position (*Figure 1C* and *Video 1*, left). In a series of experiments, we deposited extract and waited 30 min, after which we measured the shortest distance *b* of the centrosome from the boundary at steady state. This distance varied between *R*/2 and *R*, with *R* being the explant radius (*Figure 1D*). Deviation from precise centering (*b* = *R*) may be due to yolk or lipid droplets (magenta circles in *Figure 1C*) forming exclusion zones. In large explants it is conceivable that aster centering is not achieved. We also note that in some explants the single aster was by chance already positioned near the center, and no further migration occurred. To obtain more detailed insight, we analyzed the kinematics of single asters located near the boundary after cytosol deposition (*Figure 1E*). Typically, they stayed for up to 10 min (*Figure 1F*, phase I), but they always eventually migrated (*Video 1*, left). Single asters moved rapidly after separation from the boundary, with a maximum velocity of 0.05 ± 0.02 μm/s (*Figure 1F*, phase II), at around 20% of its final distance from the explant boundary (*Figure 1G*), before decelerating (*Figure 1F*, phase III) and stopping 30–35 μm from the boundary at most (*Figure 1E*).

We then measured the radial intensity profile of single asters in explants as a proxy for aster size and microtubule length (*Figure 1H*, inset). We initially focused on asters near the center of the explants. Away from the MTOC, the distribution was well approximated with a mono-exponential decay with decay length of ~12 μm (*Figure 1H*). This value agrees with the size of asters associated to telophase and early interphase nuclei of

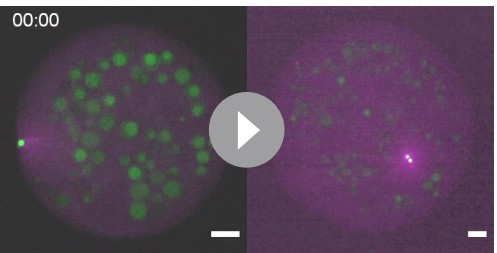

**Video 1.** Maximum intensity Z-projection from a 3D time-lapse movie of explants generated from *gnu* mutant embryos expressing RFP::β-Tubulin (magenta) and Spd2::GFP (green). The left explant contains a single aster moving away from the explant boundary, the right explant contains two separating asters. The jiggling spheres are yolk droplets. Time in min:s, scale bar 10 μm. Frame rate is 4 frames/min.

https://elifesciences.org/articles/90541/figures#video1

wildtype embryo explants (*Telley et al., 2012*). The signal drops to background level at ~30 μm, suggesting that few microtubules are longer. We also looked at the microtubule distribution in asters positioned near the explant boundary. The microtubule signal toward the explant boundary was noticeably reduced (*Figure 1—figure supplement 1*, *Figure 1*) suggesting fewer microtubules reach a length equal to the boundary distance *b*. In some samples, we observed splay of microtubules when the aster was near the explant boundary (*Figure 1I*), consistent with microtubules pushing against the boundary.

In summary, a single aster moves away from a cytoplasmic boundary and, provided sufficient space, reaches a boundary separation distance comparable to aster size. The motility displays three distinct dynamic phases: first, a very slow phase (at least along the radial axis) as they separate from the edge; second, a rapid motion away from the explant boundary; and finally, a gradually slowing down as they migrate toward the explant center (as evidenced in *Figure 1D–F*). The initial phase of separation may be due to splay of microtubules near the boundary edge (*Figure 1I*). In our movies (e.g., *Video 1*, left), we see random fluctuations in the movement of the aster and the surrounding cytoplasm while the aster is close to the explant boundary. These may be sufficient to release the aster eventually from the boundary (transition from phases I to II, *Figure 1F*). These observations are consistent with the asters generating a repulsive potential that decays to zero for distances >30 μm and is also inefficient at very short distances (<3 μm).

## Lipid droplet movements in extract are consistent with a repulsive aster potential

Embryos contain high amounts of lipid droplets and yolk granules, which serve as fiduciary markers in our explants to study hydrodynamic flow (*Monteith et al., 2016*; *Shamipour et al., 2023*). Importantly for our later results, the spatial scale of such flows can define the length scale over which the forces generated by asters act. According to the hydrodynamic pulling model, cytoplasmic dynein moves small organelles along astral microtubules toward the MTOC (*Hamaguchi and Hiramoto, 1986*). Thus, we expect small droplets and spherical organelles to occupy the space of the aster and possibly accumulate at the MTOC.

We examined the localization and movement of droplets as the aster moved through the explant (small spheres in *Video 1*). We observed an approximately circular droplet exclusion zone of ~10 μm radius (*Figure 2A, B*), which maintained during aster migration (*Figure 2B*). This contrasts with the expected observation in the hydrodynamic drag model, and is suggestive of a repulsive interaction between the aster and yolk granules, at least over distances on the order of 10 μm. Next, we quantified the mobility of these yolk granules relative to the movement of the aster. As control, we calculated the mean-squared displacement (MSD) of the lipid droplets with and without an aster present in the explant (*Figure 2C*). The droplets clearly displaced further when an aster was present. The scaling of MSD with time is indicative of the dynamic mode, $x_{ms} \sim t^{\alpha}$ . For $\alpha \approx 1$, the motion can be approximated as diffusion-like, whereas $\alpha > 1$ implies directionality in the droplet movement. With and without an aster the mode of motion was $\alpha \approx 1.5 \pm 0.1$ and $\alpha \approx 1.3 \pm 0.1$, respectively.

To better understand the dynamics, we quantified the movement of the yolk/lipid droplets relative to aster movement. The yolk granules streamed around the aster exclusion zone (*Figure 2E*) and consistently in the opposite direction, for particles in front of or behind the aster (*Figure 2F*). Again, this contrasts with the concentric movement pattern expected from the hydrodynamic pulling model. Overall, our particle motion analysis suggests a repulsion of small organelles from the aster center over a length scale of ~20 μm.

## Aster–aster interactions are consistent with a repulsive potential involving short-ranged inhibition

Some explants contained a pair of asters that separated and adopted a steady-state inter-aster distance (*Figure 3A* and *Video 1*, right). At steady-state, the aster–aster interaction must balance with the forces involved in moving each aster away from the boundary. Here, we use our quantitative measurements of aster–aster dynamics to infer an effective interaction potential that we later use to develop our theoretical model.

Our previous experiments indicate that this aster–boundary force decays with distance. From the perspective of the first aster, we may assume the second aster forms a local boundary – which is

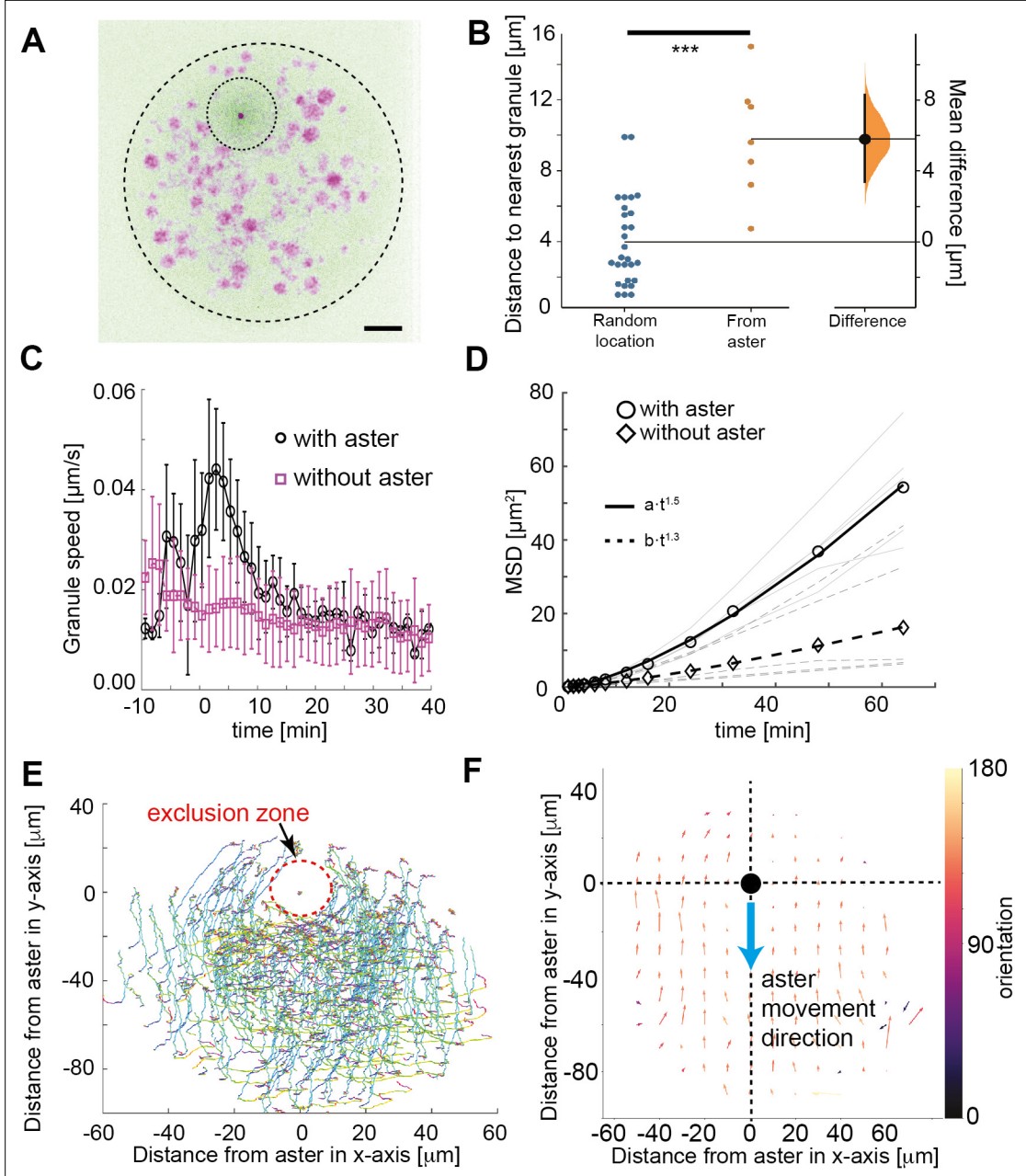

**Figure 2.** Tracking of yolk granules suggests particle displacement by repulsion. (**A**) Maximum intensity Z-projections of a single aster in an explant produced from a *gnu* mutant embryo expressing RFP::β-Tubulin (green) and Spd2::GFP (dark dot). Yolk spheres are visible in magenta due to auto-fluorescence. The dashed circle represents the explant boundary, the dotted circle highlights the droplet exclusion zone where the aster is located (**B**) Measured minimum distance between aster center (orange) or randomly generated location (blue) and nearest yolk granules when the aster was 15 μm from the boundary (Methods). ***p < 10⁻³ Mann–Witney test (*n* = 7 explants). Scale bar 10 μm (**C**) Measured yolk granule speed in the droplets with (black) and without (magenta) an aster present (*n* = 8 experiments, >100 granules tracked). Error bars denote standard deviation (SD). (**D**) Mean-squared displacement (MSD) plot of lipid droplets in the explants. Average droplet movement analyzed with (circles, solid line) and without (diamonds, dashed line) an aster present (corresponding gray and dashed gray lines show individual experiments). The continuous and the dashed line represent fits to respective models as described in the legend. (**E, F**) Velocity profile of granules relative to the coordinate system (origin) defined by the aster position, orientated such that the aster moves in the negative *y*-direction. (**E**) Shows individual granule tracks, color coded by time (light green start through to red at end). (**F**) Averaged granule movement over seven experiments, with the direction of aster movement highlighted by blue arrow. Granule movement orientation is color coded, and the length of arrows represents speed.

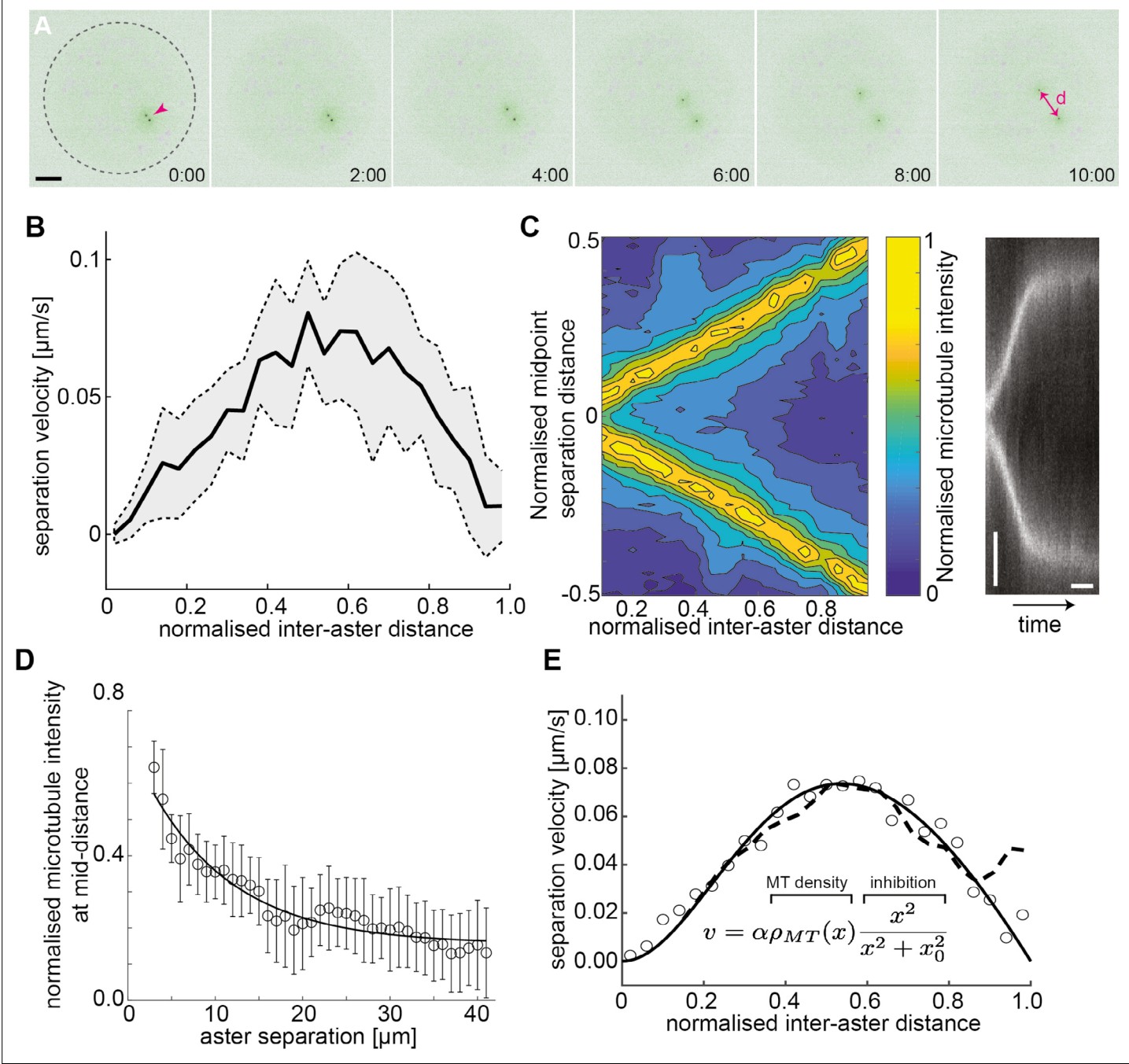

**Figure 3.** Aster–aster separation in explants depends on microtubule distribution and interactions. (**A**) Maximum intensity Z-projections of two asters (arrowhead) separating in an explant produced from *gnu* mutant embryos expressing RFP::β-Tubulin (green) and Spd2::GFP (black dots). Yolk spheres are visible in magenta due to auto-fluorescence. In the last frame, the separation distance $d$ is marked by a double arrow. The dashed circle represents the explant boundary. Scale bar, 20 µm. (**B**) Aster separation velocity as a function of normalized separation distance ($n$ = 9). For each experiment, distance is normalized by the final, steady-state separation distance. (**C**) Left: Colormap of normalized microtubule density between two separating asters. Right: Kymograph of microtubule intensity between the asters during separation. Scale bars, 2 min (horizontal) 5 µm (vertical). (**D**) Normalized microtubule intensity at the midpoint perpendicular axis between asters in function of aster separation distance. Open markers denote average values and error bars the standard deviation. Solid line represents the fitting to exponential decay ($n$ = 7). (**E**) Fitting to average separation velocity (circles) considering microtubule intensity and a short-range inhibition term (inefficient repulsion). Microtubule density was either fitted beforehand (solid line in **D**) or directly included (dashed line).

The online version of this article includes the following figure supplement(s) for figure 3:

**Figure supplement 1.** Inter-aster steady state distance and separation dynamics.

movable – and associate a similar force generation property between the first and the second aster. The resulting mechanical configuration is likely symmetric so that asters 1 and 2 are interchangeable. Force balance considerations (see Methods) provide a testable hypothesis: If the force occurring between the two asters ($F_{a \leftrightarrow a}$) or between an aster and the boundary ($F_{a \leftrightarrow b}$) have identical mechanical properties, that is, the same amplitude and length parameters, we expect the asters to be positioned at half the diameter of the explant. Thus, we measured the final distance $d$ between the centrosomes of the two asters, as well as the boundary distances $b_1$ and $b_2$ for asters 1 and 2, respectively, and the explant radius, $R$. Interestingly, we find that the two asters approximately partitioned the available space ($d = b_1 = b_2 = 2R/3$) (*Figure 3—figure supplement 1A*). From this result, we can conclude that from the viewpoint of one of the two asters, the periphery of the second aster cannot be viewed as a hard mechanical object. In other words, the steady-state force generation associated with an aster being a distance $x$ away from the periphery of the second aster is lower than the force generated between the same aster and the explant boundary at distance $x$.

Given the above observation, we predicted that the dynamics between aster pairs will differ from the single aster dynamics. Using live imaging, we tracked the dynamics and separation of aster doublets. The velocity profile of the doublets is distinct from the single aster case (*Figure 3—figure supplement 1B*). We noticed that the peak separation velocity was always near half the final aster separation distance (*Figure 3B*), independent of final separation distance. This suggests that the resulting magnitude of the forces are similar in the initial phase of separation and the eventual reaching of the equilibrium position.

Given the eccentric movement of two asters, the aster separation could be driven by overlap and sliding of astral microtubules (*Baker et al., 1993*; *Deshpande et al., 2021a*; *Lv et al., 2018*; *Vukušić et al., 2021*; *Vukušić et al., 2017*), or by mutual contact leading to repulsion by microtubules of both asters. Thus, we quantified the microtubule intensity between the separating asters (*Figure 3C*, left) and generated kymographs of the microtubule fluorescence intensity along the separation axis (*Figure 3C*, right). The intensity at half the separation distance decayed exponentially (*Figure 3D*), consistent with models of dynamic microtubule length distribution (*Howard, 2001*; *Jeune-Smith and Hess, 2010*). When aster separation ceased there was almost no detectable microtubule signal between the asters.

Viscous forces dominate at the cellular scale, and we expect that the net force causing aster separation is related to the velocity of separation, because it must balance the drag force caused by their movement through the bulk cytoplasm. For a viscous material, the velocity, $v$, of a submerged object depends on the applied force $F$: $v \approx \gamma F$, where $\gamma$ is the effective viscous drag coefficient. Naively interpreting the microtubule distribution as the resulting force profile does not match with the observed separation velocity profile. However, multiplying the microtubule distribution by a short-ranged inhibitory term, $f_{inhib} = f_0 \frac{x^2}{x_0^2 + x^2}$ ($x_0 \approx 15\mu m$), results in an excellent fit to our observed aster separation velocities (*Figure 3E*). The nature of such a short-ranged inhibition in the action of microtubules is expanded on in the Discussion.

Overall, we see that aster–aster and aster–boundary dynamics both appear to involve repulsive interactions with a degree of inhibition at very short distances. The difference in the apparent steady-state positioning of aster pairs suggests that aster–aster interactions are weaker than those between the asters and the boundary. Below, we use these results to define length scales and relative interactions strengths to simulate an effective potential between asters to explain the observed dynamics.

The two asters may mechanically interact via crosslinking of microtubule overlaps (*Bieling et al., 2010*; *Deshpande et al., 2021a*; *Lv et al., 2018*; *Subramanian et al., 2013*; *Wijeratne and Subramanian, 2018*), while astral microtubules may simply hit against the boundary interface, which acts as an immovable hard wall. However, we note that this conclusion is based on inference from the aster positions; we have not been able to quantitatively test this observation.

## Perturbations of aster interaction are consistent with a microtubule-mediated repulsive force potential

To further explore the nature of microtubule aster interaction, we performed a series of inhibitory treatments to chemically perturb the interaction. Since small-molecule inhibitors for candidate molecular motors have no effect in *Drosophila* (*Firestone et al., 2012*; *Maliga et al., 2002*), we targeted microtubules and ATPases in general. We generated explants with two asters during separation and

pulse injected a defined volume of 200 μM colchicine, which causes acute depolymerization of microtubules. Upon injection the asters stopped separating and sometimes inverted their direction of motion (*Figure 4A* and *Video 2*, right). We then tested whether ATP-dependent molecular machinery was the sole process leading to aster repulsion, by inhibiting ATP consumption with sodium azide. We injected a series of concentrations of sodium azide into explants that contained a separating pair of asters (*Video 2*, middle). Adding sodium azide decreased the initial recoil velocity (dashed lines in *Figure 4A*) and resulted in a considerable reduction in aster separation distance. However, even at very high concentrations of sodium azide, we still observed residual motion, suggesting that both ATP-driven microtubule-mediated separation and passively driven separation, for example, through entropy minimization, occur here.

Both effective pushing or pulling forces can cause aster separation and centering (*Grill et al., 2003*; *Laan et al., 2012*; *Zhu et al., 2010*). Though our above results appear more consistent with effective pushing forces, this is based on analysis of the aster dynamics and the aster positioning within the explant; that is, we have not directly tested the nature of the effective force. Pulling within the cytoplasm requires aster asymmetry (*Kimura and Kimura, 2011*; *Tanimoto et al., 2016*). Thus, we performed targeted UV photo-ablation experiments in larger explants containing one or two asters, thereby inducing shape change or inhibiting interaction of the asters (*Figure 4B*, see Methods). First, we generated ellipse-shaped ablations positioned asymmetrically around one steady-state aster, affecting microtubules on the left side more than on the right side of the aster (*Figure 4C*). If pulling on the boundary (*Grill et al., 2003*) or hydrodynamic effects from vesicle transport along microtubules (*Tanimoto et al., 2016*) drives aster motion, we expect a displacement to the right (positive) after ablation. Conversely, if the net force applied on microtubules favors pushing on the MTOC, we expect a displacement to the left (negative). Indeed, asters consistently moved to the left, supporting a dominating effect of microtubule-driven pushing (*Figure 4D* and *Video 3*). As a control, we performed the same perturbation in explants that were injected with the microtubule inhibitor colchicine (*Figure 4E*). Under this condition, asters moved very slowly to the right (positive), which is consistent with a weak hydrodynamic effect from other contractile sources (e.g., actomyosin) (*Kinneret, 2016*). We conclude that a single aster moves and positions within *Drosophila* explants by microtubule-dependent pushing force.

To challenge these conclusions, we performed two types of UV ablation in explants containing two asters (*Figure 4F*): (1) light pulses emitted along an ellipse around both asters, while they separate, to destroy microtubules in the periphery; (2) light pulses emitted along a line between the two asters, either in steady state or while separating, to destroy microtubules between asters. If forces are attractive, then ablation type 1 will stop separation while ablation type 2 will lead to an acceleration. If forces are repulsive, we predict the opposite response. We found a slight but significant acceleration for peripheral ablation in two out of three experiments (*Figure 4G*, type 1 and *Video 4*). We observed a strong deceleration or movement inversion with subsequent recovery for central ablation (*Figure 4F*, type 2) in all three experiments. Separation recovered likely because of fast re-growth of microtubules after ablation (in the range of μm/min) (*Rogers et al., 2002*). In summary, the dynamic behavior of asters in our explants is consistent with a model of radially symmetric microtubule-based repulsion.

## Simple repulsive model of aster interactions can replicate the observed aster behavior ex vivo

We formulated a physical model of aster repulsion from our experimental insights. We considered the asters as generating a radially symmetric repulsive potential, embedded within a 2D circular environment (*Figure 5A*). The repulsive potential is taken to be exponentially decaying with length scale 15 μm (based on *Figure 1H*). We also include a short-ranged inhibitory term when asters are close to the boundary or each other, of the form $\frac{x^2}{x^2+x_0^2}$, where $x_0 = 18 \mu m$ (aster–boundary) and $x_0 = 25 \mu m$ (aster–aster). To match our observations on one- and two-aster cases, the repulsion from the explant boundary was 40% larger than the aster–aster repulsion (Methods). We also include a noise term due to the inherent stochasticity in the system.

We tested whether this 2D model could replicate the observed dynamics and positioning in the one-aster case. The model reproduced the dynamics of a single aster moving away from the boundary

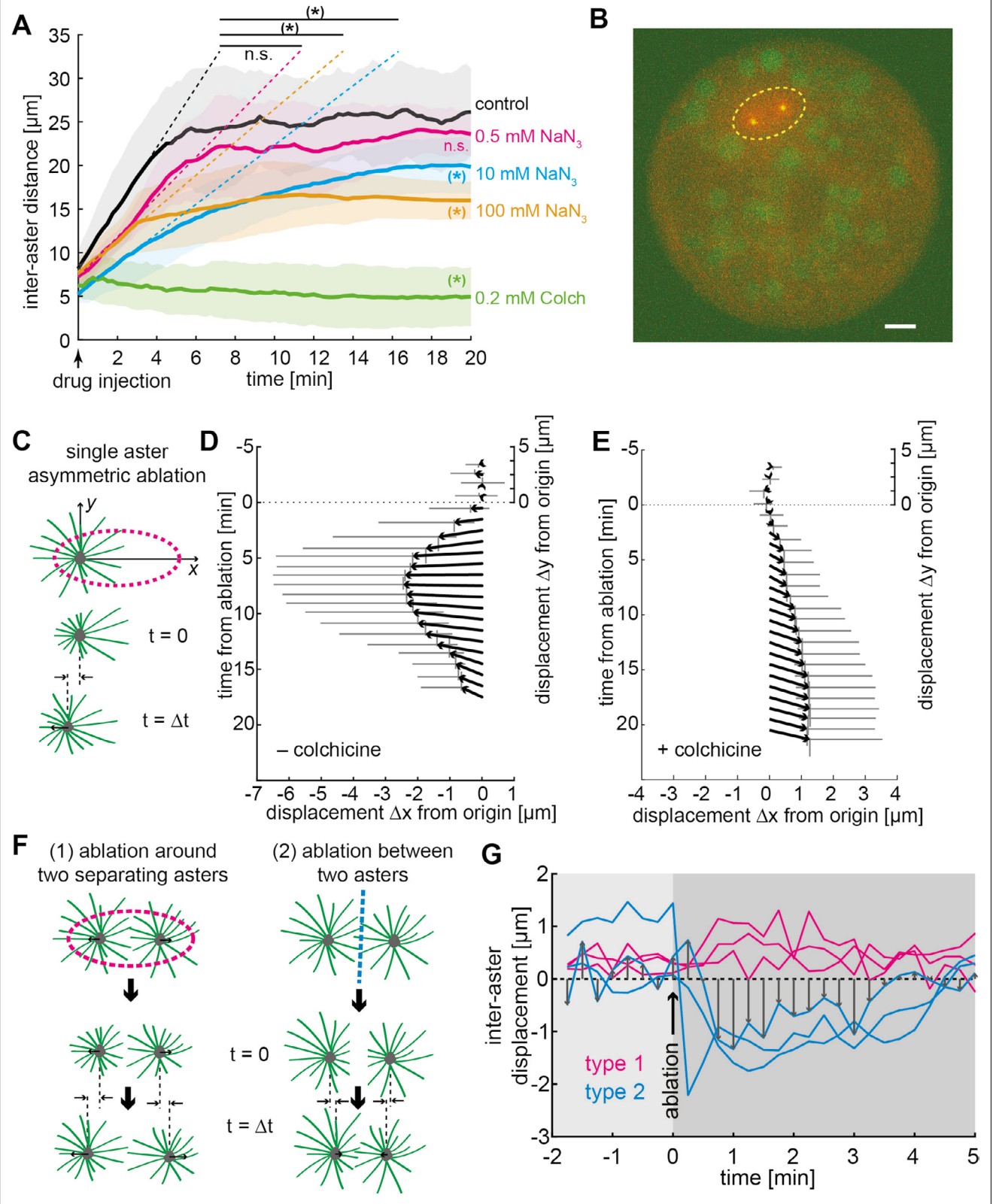

**Figure 4.** Aster positioning and separation is determined by a dominant microtubule-dependent pushing force. (**A**) Aster separation dynamics upon injection of buffer (control, $n = 3$), 0.5 mM ($n = 3$), 10 mM ($n = 4$), 100 mM ($n = 3$) sodium azide, or 0.2 mM ($n = 3$) colchicine. The * symbol denotes significance at $p < 0.05$. Gray or colored areas around average curves denote ±1 standard deviation (SD). (**B**) Sample image of an explant containing a pair of separating asters during UV laser ablation (dashed ellipse) provoking an instantaneous change of aster geometry. Scale bar 10 µm. (**C**) Schematic

*Figure 4 continued on next page*

*Figure 4 continued*

of single aster eccentric circular UV laser ablation (magenta dashed line); this ablation aims at shortening astral microtubules on the left side of the aster. $t$ = 0 min denotes ablation time. Aster displacement before and after eccentric circular ablation in explants unperturbed (**D**), $n$ = 8 or treated with colchicine (**E**), $n$ = 8. Arrows represent average displacement magnitude and direction, and vertical and horizontal gray bars denote ±1 SD of displacement in $x$ and $y$, respectively. (**F**) Explants containing two asters were perturbed by (1) ellipse ablation around both asters during separation (peripheral ablation); (2) linear ablation between two asters (central ablation). (**G**) Change of inter-aster distance (displacement) upon laser ablation (time = 0) as described in (**F**). Upon peripheral ablation, separating asters maintained or slightly accelerated their separation movement, while central ablation caused movement inversion and asters approaching each other.

(*Figure 5B*). Likewise, we can solve the 1D equation of motion stochastically for the one-aster case (Methods); the final aster position scaled with droplet size up to around 50 μm (*Figure 5C*).

We next introduced a second aster into the 2D model. We had to account for interactions between aster–boundary and aster–aster. The two-aster scenario was defined by allowing a single aster to divide, then following the positions of each sister aster. We can reproduce the experimentally observed aster dynamics, with a more symmetric speed profile as compared with the one-aster case (*Figure 5D*). Again, solving the equivalent 1D equation of motion stochastically, we found that this provides a good approximation of the aster positions (compare *Figure 5E* and *Figure 5F*).

Our model closely matches experimental observations in one- and two-aster scenarios. What about systems with greater than two asters? We previously showed that multi-aster samples form symmetric structures, for example, equilateral triangles with three asters and square-like distributions with four asters (*de-Carvalho et al., 2022*). Can our model replicate these observations? We ran our dynamic simulations with three of four initial asters randomly placed, until the asters reached equilibrium positions. In the three-aster scenario, most simulations resulted in the asters distributed such that they (approximately) formed the vertices of an equilibrium triangle. Subsequently, the asters were distributed with angle 60° between each other (*Figure 5G*). Likewise, most four-aster simulations resulted in the asters (approximately) forming the vertices of a square, with angle distribution 90° (*Figure 5H*). These results are similar to those observed experimentally (*de-Carvalho et al., 2022*). Consistent with experiment, in a small subset of simulations the four asters formed a triangle with the fourth aster positioned away from the other three (upper right inset, *Figure 5H*). In conclusion, our simple repulsive model can reproduce both the observed equilibrium aster distributions and the quantified aster dynamics in a range of scenarios.

## Discussion

We have characterized the mechanics of microtubule aster positioning using an ex vivo model of the cellular context where, naturally, hundreds of these cytoskeletal structures co-exist and define

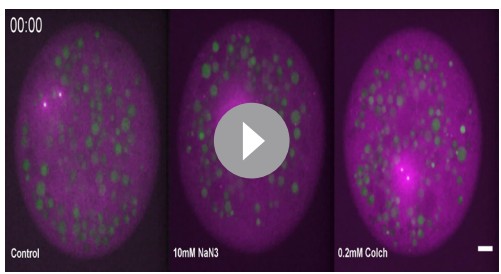

**Video 2.** Maximum intensity Z-projection from a 3D time-lapse movie of explants generated from a *gnu* mutant embryo expressing RFP::β-Tubulin (magenta) and Spd2::GFP (green), containing two separating asters, after pulse injection of solutions: control with buffer (left), 10 mM sodium azide (middle), and 0.2 mM of colchicine (right). Time in min:s, scale bar 10 μm. Frame rate is 4 frames/min.

https://elifesciences.org/articles/90541/figures#video2

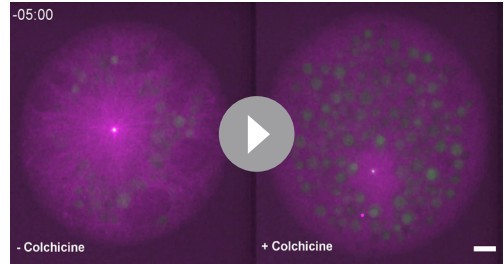

**Video 3.** Maximum intensity Z-projection from a 3D time-lapse movie of explants generated from a *gnu* mutant embryo expressing RFP::β-Tubulin (magenta) and Spd2::GFP (green) containing a single aster. The aster was allowed to equilibrate followed by an asymmetric elliptic ablation (yellow line at times 00:15 to 01:00) performed in control explants (no injection) and in explants supplemented with 0.2 mM of colchicine. Time in min:s, scale bar 10 μm. Frame rate is 4 frames/min.

https://elifesciences.org/articles/90541/figures#video3

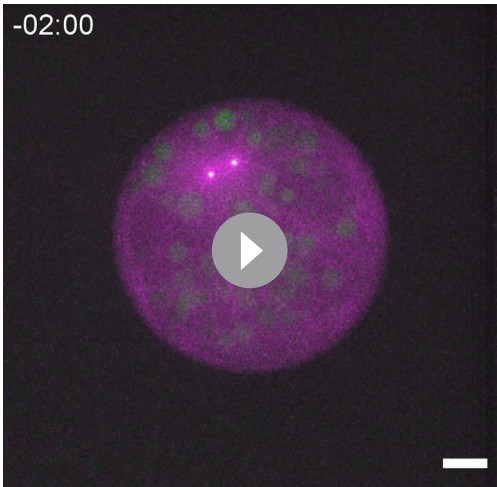

**Video 4.** Maximum intensity Z-projection from a 3D time-lapse movie of an explant containing two separating asters from a *gnu* mutant embryo expressing RFP::β-Tubulin (magenta) and Spd2::GFP (green). The elliptic ablation (yellow line from 00:15 to 00:45) was performed when asters were ~7 µm apart. Time in min:s, scale bar 10 µm. Frame rate is 4 frames/min.

https://elifesciences.org/articles/90541/figures#video4

the regular positioning of nuclei in a multinucleated cell. We generated single embryo explants from mutant *Drosophila* syncytial embryos in which mitosis is inhibited but centrosomes duplicate and divide, each giving rise to a radial microtubule array. This experimental reductionist approach of generating a proliferative aster system ex vivo has several advantages (*de-Carvalho et al., 2022*). First and foremost, it enables the study of the mechanics of single aster positioning (the 'atomic' structure) under boundary conditions, and the canonical interaction between two structures. Our experiments support a mechanical model where asters generate a radially symmetric, repulsive force potential. Given the positioning of asters relative to each other and to the boundary, we posit that the repulsive interaction toward the boundary is about twice as strong as compared to the repulsion between two asters. However, we note that we have been unable to directly measure this force. The range of action between asters is finite and productive for positioning only up to ~35 µm. We note that two asters positioned close (<3 µm) to each other, for example after centrosome disengagement, show dynamics of weak attraction. We highlight that, by design, our experiments in embryo explants resolve the canonical aster–aster interaction whereas additional cell cortex interactions and more complex boundary conditions imposed by the cell membrane may occur in the intact embryo (*Foe et al., 2000*; *Postner et al., 1992*; *Winkler et al., 2015*). Importantly, embryo explants do not reconstitute a compartmentalized f-actin cortex (*de-Carvalho et al., 2022*) as typically seen in embryos (*Foe et al., 2000*). Potentially, the boundary force in embryos may be different, with cortical attachment leading to net pulling between boundary and aster.

There has been substantial discussion over the nature of aster force potentials (*Garzon-Coral et al., 2016*; *Meaders et al., 2020*; *Minc et al., 2011*; *Mitchison et al., 2012*; *Pelletier et al., 2020*; *Sulerud et al., 2020*; *Tanimoto et al., 2018*): are they repulsive or attractive; what range do they act over; and are there different regimes of action depending on temporal or spatial constraints? Here, we show that asters derived from the syncytial embryo of *Drosophila* display a short-range repulsive potential. At very short distances, this potential tends to zero (or arguably even attractive), likely due to microtubules being unable to form linearly. A simple model can replicate our observations without requiring additional assumptions. It is worth noting that the length scales here (typically 3–10 µm between asters in the embryo) are substantially smaller than those in other model systems used to explore aster dynamics, such as the sea urchin (>50 µm, *Tanimoto et al., 2018*; *Meaders et al., 2020*). In that system, effective pulling forces generated by hydrodynamic processes dominate the aster positioning (*Tanimoto et al., 2018*).

What is the mechanism underlying the reduction in microtubule-mediated force at very short distances, both for aster–aster and aster–boundary interactions? We observed splay in the microtubule distribution near the boundary. If microtubules are generating a mechanical pushing force (i.e., like a rod being pushed against a wall), such splay would be expected. Microtubule polymerization at a boundary generates forces against the boundary in the range that can lead to buckling (*Holy et al., 1997*; *Howard, 2001*). Conversely, it is challenging to reconcile this observation with astral microtubules (and linker proteins) transmitting a local pulling force (*Grill et al., 2003*). However, when two asters are very close, there may occur molecular crosslinking both between microtubules orientated (anti-) parallel – and perpendicular – to their axis of separation. Crosslinking between the

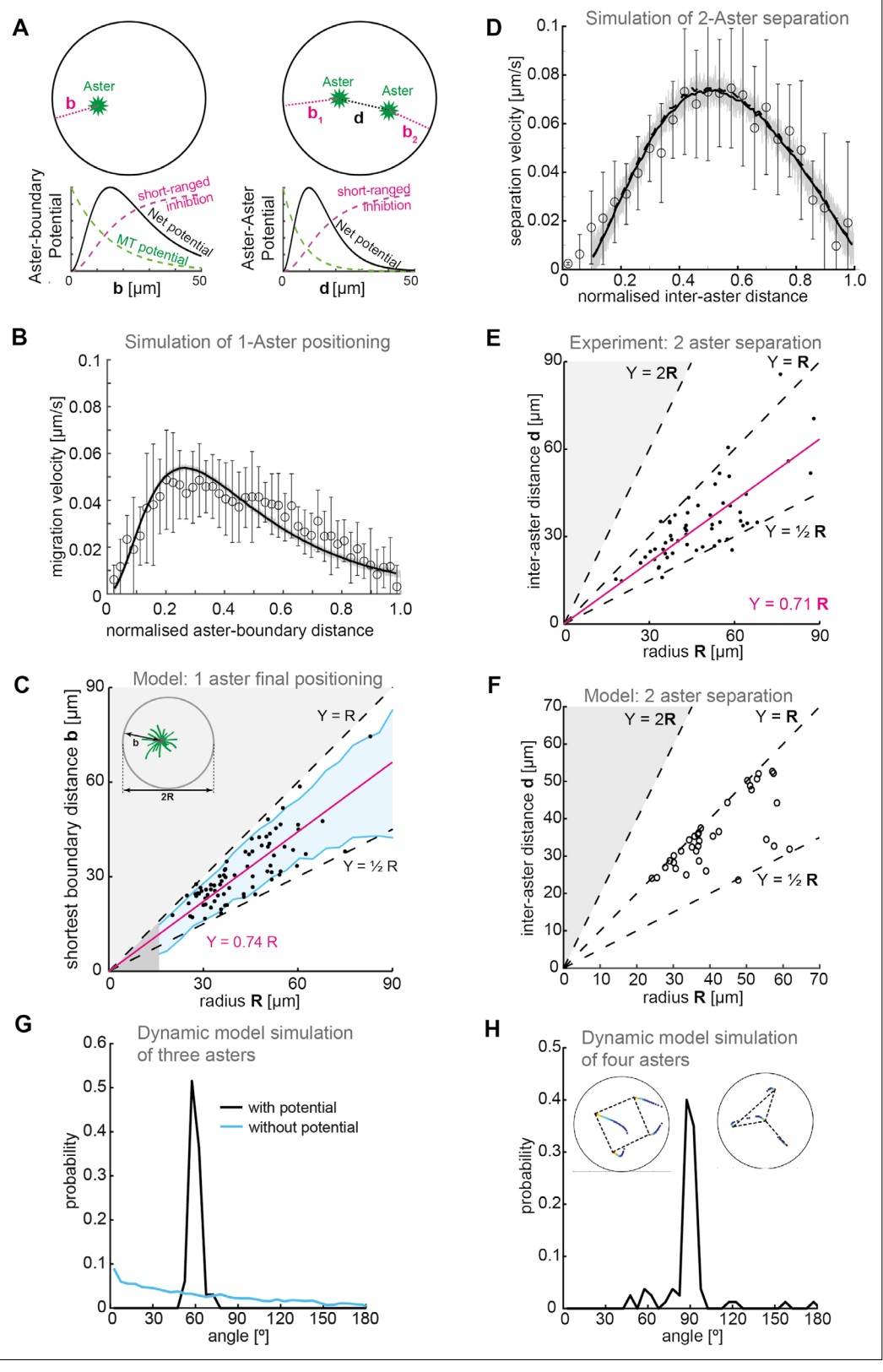

**Figure 5.** Model. (**A**) Schematic of a 2D model of aster dynamics. Bottom graphs represent cartoons of the potentials between aster–boundary and aster–aster. (**B**) Fit of 2D model to the single aster velocity profile shown in *Figure 1E*. The average model fit (black curve) is calculated from 10 simulation runs (gray lines). Error bars are experimental error (*n* = 7). (**C**) 1D stochastic model of single aster dynamics (red line = mean, blue shaded

*Figure 5 continued on next page*

*Figure 5 continued*

region ±1 standard deviation [SD]), compared to experimentally observed distribution of aster position (black dots, *Figure 1D*). (**D**) As in (**B**) but for the two-aster scenario. (**E**) Scattered plot of inter-aster distance ($d$) as a function of the radius ($R$) of explants containing two asters ($n = 54$). Most measured data points fall between the dashed lines denoting the explant radius ($Y = R$) and half of the radius ($Y = \frac{1}{2} R$). The magenta line represents the linear regression. (**F**) 1D stochastic model of two-aster dynamics where black circles denote final aster positions from simulation. (**G**) Angle distribution from aster positions in a dynamic model simulation with three asters. The simulation evolved from initially random positions, and asters robustly moved toward a triangular configuration. The peak at 60° represents equal distances between the three asters. In the absence of a repulsion potential the regularity is lost (blue line). (**H**) Angle distribution from aster positions in a dynamic model simulation with four asters. The two insets show the temporal evolution of position and the final configuration marked with dashed lines. The majority of simulations (17/20) resulted in a regular square (top left inset) with 3/20 resulting in a 'Y' configuration (top right inset).

perpendicular orientated microtubules at the periphery could act to generate a local pulling force between the asters at very short distances. As asters separate, sliding of antiparallel orientated microtubules will dominate, driving further aster separation. To test this hypothesis, super resolution microscopy may be necessary to gain sufficient spatial resolution to distinguish the relative population of parallel and perpendicular aligned microtubule bundles.

Between two asters, force inhibition may be attributed to molecular friction between microtubules (*Forth et al., 2014*). Single-molecule experiments showed that antiparallel microtubule crosslinking, and their sliding by molecular motor activity or entropic effects (*Lansky et al., 2015*), results in visco-elastic properties of the microtubule pair as a mechanical element (*Shimamoto et al., 2015*). At the level of two asters, hundreds of such elements in parallel accumulate to a 'softened' repulsion, over-coming any opposing forces on each aster (boundary constraints, viscosity of cytoplasm), leading to aster separation.

We emphasize that the molecular mechanism underlying the generation of the pushing forces remains unclear, in part due to the lack of suitable reagents for targeted perturbation of molecular motors in *Drosophila*. However, our finding that aster separation occurs despite inhibition of ATPases (e.g., motors) is both interesting and intriguing. It is unlikely due to mutual contact between microtubules growing from opposite asters since the probability for such encounters to happen and to be mechanically effective is extremely low. In our view, this 'passive' repulsion force is likely caused by entropy-driven microtubule sliding by ATP-independent crosslinkers (*Forth et al., 2014*). If possible, targeted inhibition of candidate microtubule-associated proteins should give further insight.

## Materials and methods

**Key resources table**

| Reagent type (species) or resource | Designation | Source or reference | Identifiers | Additional information |
|---|---|---|---|---|
| Genetic reagent *Drosophila melanogaster* | w*; pUbq>RFP::β2-Tubulin; + | Gift from Yoshihiro Inoue (*Inoue et al., 2004*) | | |
| Genetic reagent *D. melanogaster* | w¹¹¹⁸; pUbq>Spd2::GFP; + | Gift from M. Bettencourt-Dias (*Nabais et al., 2021*) | | |
| Genetic reagent *D. melanogaster* | *gnu³⁰⁵* | Bloomington *Drosophila* Stock Center | Stock no. 3321 (discontinued), Bloomington | w*; +; gnu³⁰⁵/TM3 |
| Genetic reagent *D. melanogaster* | *gnu³⁷⁷⁰ᴬ* | Bloomington *Drosophila* Stock Center | Stock no. 38440 (discontinued), Bloomington | w*; +; gnuᶻ³⁻³⁷⁷⁰ᴬ/TM3 |
| Genetic reagent *D. melanogaster* | w¹¹¹⁸; CyO/Sco; MKRS/TM6B | Bloomington *Drosophila* Stock Center | Stock no. 3703, Bloomington | |

*Continued on next page*

*Continued*

| Reagent type (species) or resource | Designation | Source or reference | Identifiers | Additional information |
|---|---|---|---|---|
| Genetic reagent *D. melanogaster* | w\*; pUbq >Spd2::GFP, pUbq >RFP::β2-Tubulin; gnu<sup>305</sup>/gnu<sup>Z3-3770A</sup> | This study | | See Materials and methods |
| Chemical compound, drug | Halocarbon oil | Voltalef Oil 10 S, Arkema Inc | | |
| Chemical compound, drug | Colchicine | Sigma-Aldrich | CAS 64-86-8, product no. C9754 | ≥95% (HPLC), powder |
| Chemical compound, drug | $NaN_3$ | Sigma-Aldrich | CAS 26628-22-8, product no. 71290 | Sodium Azide purum 99% |
| Software, algorithm | TrackmateJ | *Schindelin et al., 2012*; *Tinevez et al., 2017* | | |
| Software, algorithm | Matlab | MathWorks Inc | RRID:SCR_001622 | |

## Fly strains

Flies expressing a fluorescent reporter for microtubules and the centrosome were generated by recombination of the genotypes w\*; pUbq >RFP::β2-Tubulin; + (*Inoue et al., 2004*) and w<sup>1118</sup>; pUbq >Spd2::GFP; + (provided by M. Bettencourt Dias, IGC, Portugal). Two different mutants of giant nucleus (*gnu*), namely w\*; +; gnu<sup>305</sup>/TM3 (discontinued stock no. 3321; Bloomington) and w\*; +; gnu<sup>Z3-3770A</sup>/TM3 (discontinued stock no. 38440; Bloomington), were each balanced with w<sup>1118</sup>; CyO/Sco; MKRS/TM6B (stock no. 3703, Bloomington). Above-described recombined line on the second chromosome were individually crossed with gnu mutants and kept as balanced stocks. Finally, trans-heterozygous were generated for gnu<sup>305</sup>/gnu<sup>Z3-3770A</sup> mutants, whereby only flies homozygous for the fluorescent reporters on the second chromosome were selected for increased signal collection during live microscopy. These trans-heterozygotes laid fertilized eggs which undergo several embryonic rounds of chromatin replication and centrosome duplication, allowing for the study and quantification of asters at the embryo cortex.

## Embryo collection and sample preparation

We followed established procedures of fly husbandry (*Schubiger and Edgar, 1994*), keeping flies at 25°C under 50–60% humidity. For embryo collections, young adult flies were transferred to a cage coupled to an apple juice agar plate. After two to three rounds of egg laying synchronization, developing embryos were collected every 30–60 min. In the case of *gnu* mutants, embryos were collected at different time intervals, ranging from 30 min up to 4 hr. Embryos were dechorionated by short immersion in 7% sodium hypochlorite solution (VWR). After extensive rinsing with water, embryos were aligned and immobilized in a thin strip of heptane glue placed on 22 × 22 mm coverslips, and covered with halocarbon oil (Voltalef 10 S, Arkema).

## Microscopy

Time-lapse acquisitions were conducted on a Nikon Eclipse Ti-E microscope equipped with a Yokogawa CSU-W Spinning Disk confocal scanner and a piezoelectric stage (737.2SL, Physik Instrumente). For embryo imaging, 15 μm (31 planes) Z-series stacks were acquired every 15 s (wildtype, if not states else) or 30 s (*gnu* mutant), using a Plan Fluor 40 × 1.3 NA oil immersion objective, the 488 and 561 nm laser lines, and an Andor Zyla 4.2 sCMOS camera to acquire images. For explants up to 100 μm in diameter, we used a Plan Apo VC 60 × 1.2 NA water immersion objective with ×2 post-magnification and an Andor iXon3 888 EMCCD camera. When needed, the Andor Zyla 4.2 sCMOS camera was selected to acquire a ×2 wider field of view with the same spatial resolution or, alternatively, the Apo $\lambda$ S LWD 40 × 1.15 NA water immersion objective. For acquisition in explants, the frame rate was 15 s for *gnu* mutant 30 s for wildtype embryo explants.

## Single embryo explant assay

Embryo extractions were performed as previously described (*de-Carvalho et al., 2018*; *Telley et al., 2013*). Briefly, cytosol from wildtype embryos was extracted by puncturing the vitelline membrane

with a sharp glass micropipette and flow control by operating a bi-directional syringe pump. Small explants of cytosol (in the picolitre range) were deposited on poly-L-lysine coated glass surface under halocarbon oil. Time-lapse acquisitions typically started in late interphase or prophase. In the case of *gnu* mutant embryos, most extractions were performed when few centrosomes (between 5 and 40) were visible at the anterior–lateral cortex. Repeated use of the same extraction micropipette is not recommended. Explants from wildtype embryos initially containing a single nucleus were selected for time-lapse imaging of subsequent mitotic divisions. Explants from *gnu* mutants initially containing a single free aster near oil interface or two free asters close by were selected for time-lapse imaging of aster separation. All experiments were conducted at 25 ± 1°C.

## Pharmacological perturbation of embryo explants

Pharmacological perturbations were performed by adding different drugs (colchicine at 0.2 mM, sodium azide at 0.5, 10, or 100 mM) diluted in cytoplasm-compatible buffer (50 mM 4-(2-hydroxy ethyl)-1-piperazineethanesulfonic acid [HEPES], pH 7.8, 100 mM KCl, 1 mM $MgCl_2$). Solutions were directly administrated to the explants using a fine pipette (pulled using a Narishige PC-100 Puller with a: two-step (69% + 55%) heating protocol and with 4 mm drop length) connected to an Eppendorf FemtoJet 4i pump. The final drug dilution in the explants was approximately 1:10 (solution:cytosol). Buffer injections were conducted as control.

## Laser ablation system

The laser ablation system was implemented by IAT on the confocal spinning-disk microscope described above. The system, the validation and experimental controls were described in several past studies (*Telley et al., 2012*; *de-Carvalho et al., 2022*; *Milas et al., 2023*). A Crylas FTSS-355-Q pulsed laser emitting 355 nm, 1.1-ns pulses, 15-µJ pulse energy at 1 kHz was aligned with a beam expander (16×), a scan head (SCANcube 7, Scanlab, Germany) coupled to an f-theta lens (f = 56 mm, anti-reflection coating for 340–370 nm, SCANLAB AG, Germany). The focus point of the f-theta lens was aligned to be parfocal to the focal plane of the objective, using a tube lens (f = 200 mm, Ø = 30 mm, 355 nm AR coated, OWIS, Germany) and a dichroic mirror (T387 DCLP, Chroma) in the upper stage filter wheel. Any scattered light was blocked at the emission side with a RazorEdge LP 355 dichroic mirror OD6 @ 355 nm (Chroma). The system was controlled with homemade journals for Metamorph software (Molecular Devices Inc). The optimal laser power was set to ensure microtubule ablation while avoiding thermal expansion of cytoplasm, with post-ablation microtubule signal recovery matching known polymerization dynamics. This combination of conditions proved to be efficient at ablating target structures beyond fluorophore bleaching. In explants containing a single aster, astral microtubules were asymmetrically ablated by positioning an ellipsoid off-center (21.7 by 10.8 µm, four times, 15-s interval, 0.54-µm step, laser power: 25%) (*Figure 4B*). In explants containing two asters, astral microtubules were ablated using an ellipsoid (21.7 by 10.8 µm, three times, 15-s interval, 0.54-µm step, laser power: 10–15%) roughly centered at the mid-point between the two asters, while inter-polar microtubules were ablated using linear ablations (21.7 µm, three times, 15-s interval, 0.54-µm step, laser power: 10–15%) perpendicular to the axis connecting the asters (*Figure 4F*).

## Simple one-dimensional model of one- and two-aster positioning

If we consider each aster to generate a force potential that follows its microtubule distribution, then $F(x) = F_0 e^{-\frac{x}{\lambda}}$ where $x$ is the radial distance from the explant boundary and $\lambda$ is the decay length of the microtubule distribution. For the one-aster scenario, the steady-state solution ($F = 0$) corresponds to the aster positioning at the center of the explant (assuming the explant size is small enough that there is non-zero force at the boundaries). For the two-aster scenario, assuming asters push on each other just like they push on the boundary, we have $F_1(x_1) = F_0 e^{-\frac{x_1}{\lambda}} - F_0 e^{-(x_2 - x_1)/\lambda}$ and $F_2(x_2) = -F_0 e^{-\frac{(2R - x_2)}{\lambda}} + F_0 e^{-(x_2 - x)/\lambda}$, where $0 < x_1 < x_2 < 2R$ with $R$ being the explant radius. The net force is zero when $x_1 = R/2$ and $x_2 = 3R/2$.

## Two-dimensional dynamic model of aster interactions

The cytoplasm is viscous. For a viscous material, the velocity, $v$, of an object is dependent on the applied force $F$: $v \approx \gamma F$, where $\gamma$ is the effective viscous drag coefficient. In our simple dynamic model implemented in Matlab we consider $\gamma = 1$ and isolated asters with a circularly symmetric force

potential described by $f(r) = f_{slip}(r) \times \rho_{MT}(r)$, where $r$ is the distance from the aster center (centrosome), $f_{slip} = f_0 \frac{r^2}{x_0^2 + r^2}$ ($x_0 \approx 15\,\mu m$) and $\rho_{MT}(r)$ represents the distribution of microtubules from the aster. We incorporate $f_{slip}$ to account for the reduced apparent microtubule force generation at short distances. For simplicity, we take the same characteristic distance $x_0$ for both aster–boundary and aster–aster interaction. To account for boundary conditions, we introduce a mirror charge outside the circle for each aster. We solve $v \approx \gamma F$ by the Euler method in Matlab (the equations are not highly non-linear so this approach works well and is fast).

For single asters, we only consider interactions between the wall and aster. We take $\rho_{MT}(r) = e^{-r/\lambda}$, with $\lambda = 15\mu m$ and $f_0 = 0.007$ and $r$ is the perpendicular aster–wall separation. We also include a 'noise' term, $\delta f = 0.0005$. So, $\overrightarrow{f}(r) = r f_0 \frac{r^2}{x_0^2 + r^2} e^{-r/\lambda} + r_{ran}\delta f$, where $r$ is the unit vector between aster and wall, and $r_{ran}$ is a random unit vector generated at each time iteration. For two asters, the force is given by $\overrightarrow{f}(r) = r f_{aster-wall}(r) + x f_{aster-aster}(x) + r_{ran}\delta f$, where $x$ is the distance between the two asters and $x$ the unit vector between them. $f_{aster-wall}(r)$ is the same as for the one aster scenario. We take $f_{aster-aster}(x) = f_1 \frac{x^2}{x_0^2 + x^2} e^{-x/\lambda_{aster}}$, $f_1 = 0.005$ and $\lambda_{aster} = 12\mu m$.

Considering the aster–aster separation, we assumed the aster pair initially separated by 2 µm and centered within the in silico explant space. For the single aster case, we initialized the aster 1 µm from the boundary. We always considered a system of radius 40 µm. Simulations were run until the aster position reached a steady-state and angles between asters were measured at the last time point.

## One dimensional dynamic model of aster interactions

We solved the above equations of motion in one-dimension using Matlab's built in stochastic PDE solver (sde), **Figure 5C, F**. For each condition we ran 3000 simulations. For the two-aster scenario, we considered asters placed initially either side of the midpoint ($x = L/2$).

## Analysis of free asters in explants – distance distributions

Distance between asters and from aster to the boundary were obtained in explants at steady state, that is where asters did not move anymore (usually 30–45 min after explant deposition). The inter-aster distance was determined as Euclidean distance in 3D. We defined the *boundary distance* ($b$, $b_1$, $b_2$) as the shortest distance from the aster to the interface between glass, oil, and cytosol, determined manually using the Fiji measurement tools (at a precision of ± 0.5 µm). To determine the explant boundary on the glass (approximated with a circle of radius $R$), maximum intensity projections of both fluorescence emission channels was assessed to trace the interface between the glass, oil, and cytosol. For larger explants with high aspect ratio – a quasi-2D situation – the definition of *boundary distance* served as good approximation for a boundary in two dimensions. However, in small explants where the aspect ratio is not as high, two asters sometimes aligned considerably in the third dimension. In these cases, the definition for *boundary distance* led to an underestimation of the maximum projected inter-aster distance $M = 2R - b_1 - b_2$; it becomes a geometric problem in 3D and the longest dimension is not necessarily in the plane of the glass–explant interface. This is evident for some data points in small explants (yellow dots in **Figure 5E**).

## Analysis of free asters in explants – dynamics

The coordinates of free asters were obtained by applying a Gaussian blur filter (radius: 1–2 pixels) and using the plugin TrackMate v3.5.1 of Fiji ImageJ (**Schindelin et al., 2012**; **Tinevez et al., 2017**). The coordinates of detected spots were imported into Matlab for assignment and distance calculation similarly as mentioned above. The instant relative velocity was calculated using the formula: $v_i = \frac{d_{i+1} - d_{i-1}}{t_{i+1} - t_{i-1}}$, where $d$ is the 3D Euclidian distance and $t$ is time in the flanking time points of the measure point. For unperturbed experiments, data were normalized to the maximum distance achieved in the separated phase to correct for scaling effect during splitting dynamics (**Figure 1G**).

To analyze the movement of yolk granules, we performed a similar analysis using Fiji TrackMateJ (**Schindelin et al., 2012**; **Tinevez et al., 2017**). Seven extracts were analyzed with an aster present, with over 100 individual tracks of granules. MSD was then extracted across the entire time course of imaging. Similar analysis was performed in extracts without an aster. Curves (**Figure 2D**) were fitted using the 'fit' function in Matlab. Fitting the general function $a \cdot t^c$ gives a best fit for $c = 1.3 \pm 0.1$ and $1.5 \pm 0.1$ for the one aster and no aster data, respectively.

To calculate the exclusion size around each aster as it moves through the explant, we considered asters when they were 15 μm from the boundary edge. At this point, the distance to the nearest granule was measured in Fɪᴊɪ from seven explants. Furthermore, we generated four random coordinates within each explant using Matlab and measured the distance to the nearest granules from that random coordinate (in situations where the random position overlapped with a granules, a new position was randomly generated), giving 28 random locations sampled across seven explants. *Figure 2B* shows that the exclusion zone around an aster is significantly larger than the likely separation given the random location of droplets. Statistical analysis performed using https://www.estimationstats.com/.

## Microtubule profile quantification

For single asters (*Figure 1F*), we quantified the microtubule intensity using the intensity of the RFP::β-Tubulin signal. Taking the point when asters were either 5 or 20 μm from the explant boundary, we used Fɪᴊɪ to measure the microtubule intensity along a 10-μm straight line from the edge and through the aster. The line had a width of 2 μm. For each experiment, we normalized the total intensity by the maximum measured value and then binned the data in 0.2 μm bins. Hence, the recorded intensity does not reach one, and the mean intensity only reaches a maximum around 0.8 as the maximum value does not occur at the same position. Similar analysis was performed for the scenario with two asters (*Figure 3D*). In this case, the centroids of the asters were used to define a straight line along which the microtubule intensity was measured throughout the process of aster separation. From this straight line between the asters, we also generated the kymograph shown in *Figure 3C*, right.

## Analysis of free asters in explants – perturbations with drugs and UV ablation

For comparison between control and perturbation experiments, data were time aligned to the perturbation time point ($t = 0$) and plotted as average ± standard deviation (SD) from at least three replicates for each condition. The change of inter-aster distance during the first 3 s after drug injection was estimated by linear regression assuming normally distributed noise, and the confidence interval of the estimated slope served as test statistic for differences between control and perturbation. Differences in final, steady-state inter-aster distance were tested by comparing the pools of distances from the last 3 s (=12 frames), using Wilcoxon signed-rank test. A significance level of 0.05 was defined prior to testing. In the case of UV ablations, the position of the aster five frames before ablation was defined as coordinate origin. The two main axes of the ellipsoid, along which the pulsed ablation was performed, defined the cartesian coordinate system. A displacement vector of the current aster position relative to the origin was calculated for each time point. The mean and SD of axial ($\Delta x$) and lateral ($\Delta y$) displacement was plotted in time (*Figure 4C, D*).

## Acknowledgements

We thank members of the IAT and TES labs for fruitful discussions, and Virgile Viasnoff, Gianluca Grenci, Tetsuya Hiraiwa, Jacques Prost, Nenad Pavin, and Thomas Surrey for constructive comments and feedback. We thank Sophie Theis for generating *Figure 2B*. We thank the staff of the Fly Facility, the Advanced Imaging Facility (AIF), and the Technical Support Service at the Instituto Gulbenkian de Ciência (IGC). Transgenic fly stocks were obtained from Bloomington *Drosophila* Stock Center (NIH). We acknowledge financial support from the Human Frontiers Science Program (HFSP), awarded to IAT and TES and supporting JC and ST (RGY0083/2016). IAT acknowledges Fundação Calouste Gulbenkian (FCG), Fundação para Ciência e a Tecnologia for grant (IF/00082/2013); the European Commission grant FP7-PEOPLE-2013-CIG (N° 818743); LISBOA-01-0145-FEDER-007654 supporting IGC's core operation; LISBOA-01-7460145-FEDER-022170 (Congento) supporting the Fly Facility; PPBI-POCI-01-0145-FEDER-022122 supporting the AIF, all co-financed by FCT (Portugal) and Lisboa Regional Operational Program (Lisboa2020) under the PORTUGAL2020 Partnership Agreement (European Regional Development Fund). TES acknowledges funding from BBSRC (grant no. BB/W006944/1).

## Additional information

### Funding

| Funder | Grant reference number | Author |
|---|---|---|
| Human Frontier Science Program | RGY0083/2016 | Timothy E Saunders Ivo A Telley |
| Fundacao para Ciencia e a Tecnologia | IF/00082/2013 | Ivo A Telley |
| European Commission | FP7-PEOPLE-2013-CIG | Ivo A Telley |
| Biotechnology and Biological Sciences Research Council | BB/W006944/1 | Timothy E Saunders |

The funders had no role in study design, data collection, and interpretation, or the decision to submit the work for publication. Open access funding provided by Max Planck Society.

### Author contributions

Jorge de-Carvalho, Conceptualization, Data curation, Formal analysis, Investigation, Methodology, Writing – original draft; Sham Tlili, Conceptualization, Software, Formal analysis, Investigation, Methodology; Timothy E Saunders, Conceptualization, Software, Formal analysis, Supervision, Visualization, Methodology, Writing – original draft, Project administration, Writing – review and editing; Ivo A Telley, Conceptualization, Resources, Data curation, Formal analysis, Supervision, Funding acquisition, Investigation, Visualization, Methodology, Writing – original draft, Project administration, Writing – review and editing

### Author ORCIDs

Timothy E Saunders ⓘ https://orcid.org/0000-0001-5755-0060
Ivo A Telley ⓘ http://orcid.org/0000-0003-4444-1046

Reviewer #1 (Public Review): https://doi.org/10.7554/eLife.90541.3.sa1
Reviewer #2 (Public Review): https://doi.org/10.7554/eLife.90541.3.sa2
Author Response https://doi.org/10.7554/eLife.90541.3.sa3

## Additional files

### Supplementary files

• MDAR checklist

### Data availability

All original data are available on a data repository of Zenodo (CERN), https://doi.org/10.5281/zenodo.8090780. Codes are available on GitHub (copy archived at *Tim Saunders Lab, 2023*).

The following dataset was generated:

| Author(s) | Year | Dataset title | Dataset URL | Database and Identifier |
|---|---|---|---|---|
| de-Cavlho J, Tlili S, Saunders TE, Telley IA | 2023 | The positioning mechanics of microtubule asters in *Drosophila* embryo explants | https://zenodo.org/records/8090780 | Zenodo, 10.5281/zenodo.8090780 |

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
