## [Editor Report · eLife assessment]

This manuscript utilizes a *Drosophila* explant system and modeling to provide **important** insights into the mechanism of microtubule aster positioning. Although the intellectual framework of aster positioning has been worked out by the same authors in their previous work, this study provides additional **solid** evidence to solidify their model.

---

## [Referee Report · Reviewer #1 (Public Review)]

Summary:

This paper addresses the mechanisms positioning microtubule asters in *Drosophila* explants. Taking advantage of a genetic mutant, blocking the cell cycle in early embryos, the authors generate embryos with centrosomes detached from nuclei and then study the positioning mechanisms of such asters in explants. They conclude that asters interact via pushing forces. While this is an artificial system, understanding the mechanics of asters positioning, in particular, whether forces are pushing or pulling is an important one.

Strengths:

The major strength of this paper is the series of laser cutting experiments supporting that asters position via pushing forces acting both on the boundary (see below for a relevant comment) and between asters. The combination of imaging, data analysis and mathematical modeling is also powerful.

Weaknesses:

This paper has overlap in the conclusions with a previous paper from the same authors, so its impact is reduced. In Figure 2, the tracking of fluid flows is hard to see and better quantifications/analyses would lead to stronger conclusions. In Figure 4, it is not clear that the acceleration is significant and no statistical test is provided or described, as far as I can tell.

---

## [Referee Report · Reviewer #2 (Public Review)]

Summary:

The aster, consisting of microtubules, plays important roles in spindle positioning and the determination of the cleavage site in animals. The mechanics of aster movement and positioning have been extensively studied in several cell types. However, there is no unified biophysical model, as different mechanisms appear to predominate in different model systems. In the present manuscript, the authors studied aster positioning mechanics in the *Drosophila* syncytial embryo, in which short-ranged aster repulsion generates a separation force. Taking advantage of the ex vivo system developed by the group and the fly gnu mutant, in which the nuclear number can be minimized, the authors performed time-lapse observations of single asters and multiple asters in the explant. The observed aster dynamics were interpreted by building a mathematical model dealing with forces. They found that aster dissociation from the boundary depends on the microtubule pushing force. Additionally, laser ablation targeting two separating asters showed that aster-aster separation is also mediated by the microtubule pushing force. Furthermore, they built a simulation model based on the experimental results, which reproduced aster movement in the explant under various conditions. Notably, the actual aster dynamics were best reproduced in the model by including a short-ranged inhibitory term when asters are close to the boundary or each other.

Strengths:

This study reveals a unique aster positioning mechanics in the syncytial embryo explant, which leads to an understanding of the mechanism underlying the positioning of multiple asters associated with nuclei in the embryo. The use of explants enabled accurate measurement of aster motility and, therefore, the construction of a quantitative model. This is a notable achievement.

Weaknesses:

The main conclusion that aster repulsion predominates in this system has already been drawn by the same authors in their recent study (de-Carvalho et al., Development, 2022). Therefore, the conceptual advance in the current study is marginal. The molecular mechanisms underlying aster repulsion remain unexplored since the authors were unable to identify specific factor(s) responsible for aster repulsion in the explant.

Specific suggestions on the original manuscript:

Microtubules should be visualized more clearly (either in live or fixed samples). This is particularly important in Figure 4E and Video 4 (laser ablation experiment to create asymmetric asters).

Comments on the revised manuscript:

Despite my suggestion, the authors did not provide evidence confirming the actual ablation of microtubules in the specified target region. The authors argue, "Given our controls and previous experience, we are confident we are ablating the microtubules." Then, at the very least, the authors should describe (in Materials and Methods) the "controls" they employed and provide a citation to the previous study where proper ablation was validated using the same laser settings. Otherwise, readers might not be convinced of the authors' claim.

---

## [Author Response]

The following is the authors’ response to the original reviews.

**Reviewer 1**
Strengths:The major strength of this paper is the series of laser cutting experiments supporting that asters position via pushing forces acting both on the boundary (see below for a relevant comment) and between asters. The combination of imaging, data analysis and mathematical modeling is also powerful.

Author Response: We thank the Reviewer for the positive comments, especially in recognising the power of our quantitative approaches.

Weaknesses:This paper has weaknesses, mainly in the presentation but also in the quality of the data which do not always support the conclusions satisfactorily (this might in part be a presentation issue).

Author Response>: We address these concerns below.

My overall suggestion for the authors is to explain better the motivation and interpretation of their experiments and also to remove some of the observations which seem to be there because they could be done rather than because they add to the main message of the paper, which I find straightforward, valuable and supported by the data in Figure 4.

Author Response: We have extended the motivation of the study in the Introduction, and at the beginning of appropriate Results sections. We better motivate the force potential and especially the key results from Figure 4. We outline specific changes below.

In Figure 2, it is difficult for me to understand what is being tracked. I believe that the authors track the yolk granules (visible as large green blobs) and not lipid droplets. There is some confusion between the text, legends and methods so I could not tell. If the authors are tracking yolk granules as a proxy for hydrodynamics flows it seems appropriate to cite previous papers that have used and verified these methods. More notably, this figure is somewhat disconnected with the rest of the paper. I find the analysis interesting in principle but would urge the authors to propose some interpretation of the experiments in the context of their big-picture message. At this point, I cannot understand what the Figure adds.

Author Response: Indeed, we track the yolk droplets that move around the aster. In the extraction protocol, we likely get a mixture of lipid droplets and yolk granules; this is due to the extraction procedure involving shear forces within the pipette. We are not certain about the exact nature of these droplets, but they are likely to a large extent yolk. We have clarified the terminology in the text, the legend and methods section. In this figure, we now show that the droplets do not move towards the aster center as the hydrodynamic pulling model would suggest. Instead, they appear to passively respond to a repulsive force, that results in them streaming around the aster. We have added additional panels to the figure that illustrates the directionality of yolk granule movements (lines 159-164). We agree with the Reviewer that the context could have been clarified. The role of fluid flows in biological systems is, as the Reviewer highlights, well studied. We have added additional contextualisa8on in the text (lines 140-146). We also motivate more clearly the figure, as it provides evidence that the asters generate forces over 20µm scale (lines 159-164). This is highly relevant for one of the paper’s main conclusions – that the *Drosophila* blastocyst asters generate pushing forces that enable regular packing.

In Figure 3, it is not surprising that the aster-aster interactions are different from interactions with the boundary which is likely more rigid. It is also hard to understand why the force and thus velocity should scale as microtubule length. This Figure should be better conceptualized. I think that it becomes clear at the end of the paper that the authors are trying to derive an effective potential to use in a mathematical model in Figure 5 to test their hypotheses. I think that should be told from the start, so a reader understands why these experiments are being shown.

Author Response: We don’t claim that the force scales with microtubule length on a single microtubule. However, at larger distances from the aster, the microtubule density decreases, and hence the effective force decreases.

The Reviewer is correct that we use these results to motivate our effective potential. We have brought this motivation forward in the manuscript to guide the reader (lines 169-171) and included a further note at the end of the section (lines 216-218).

The experiments in Figure 4 are very nice in suppor8ng a pushing model. However, it would help if the authors could speculate what the single aster is pushing against in this experiment. The experiments reported in Figure 1 seemed to suggest that the aster mainly pushed against the boundary. In the experiments in Figure 4 do the individual asters touch the boundary on both sides? I think that readers need more information on what the extract looks like for those experiments.

Author Response: We now include an additional panel B in Figure 4– that shows an example of an explant during aster ablation. The distance between asters is typically less than the distance to the explant boundary. Boundary effects likely play a small role in the aster-aster separation, in terms of potentially determining the axis of separation. However, the separation of asters occurs along a straight line for a substan8al period (>1 min) of separation; if boundary effects were more dominant, we may expect to see curving of the aster-aster separation trajectories as they also receive feedback from the boundary.

Figure 4F could use some statistics. I doubt that the acceleration in the pink curves would be significant. I believe that the decelera8on is and that is probably the most crucial result. Since the authors present only 3 asters pairs it is important to be sure that these conclusions are solid.

Author Response: We agree with the Reviewer. These experiments are challenging to do, as they require carefully controlled conditions. In two out of three experiments we see significant increase in acceleration in the pink curves. Of course, the interpretation of this must be caveated as our experimental number is low. These details are now provided in the revision (lines 263267).

**Reviewer 2**
Strengths:This study reveals a unique aster positioning mechanics in the syncytial embryo explant, which leads to an understanding of the mechanism underlying the positioning of multiple asters associated with nuclei in the embryo. The use of explants enabled accurate measurement of aster motility and, therefore, the construc8on of a quantitative model. This is a notable achievement.

Author Response: We thank the Reviewer for their review, and in highlighting how our quantitative model is a clear step forward in our understanding of aster dynamics.

Weaknesses:The main conclusion that aster repulsion predominates in this system has already been drawn by the same authors in their recent study (de-Carvalho et al., Development, 2022). As the present work provides additional support to the previous study using different experimental system, the authors should emphasize that the present manuscripts adds to it (but the conceptual novelty is limited).

Author Response: While this study is related to the previous work, there are major differences. First, here we quantitatively assess aster dynamics within a “clean” system. Such accurate measurements are not possible in vivo currently. Further, experiments like laser ablation are much better defined within the explant system. We do recognise more clearly the previous work in the Introduc8on and lines 291-293, 299-300. Combined, with the different perspectives provided in these papers on the problem of aster positioning in syncytia, we believe these papers provide new and well-supported insights.

The molecular mechanisms underlying aster repulsion remain unexplored since the authors were unable to identify specific factor(s) responsible for aster repulsion in the explant.

Author Response: Given that the nature of the aster dynamics were not previously characterised, our work presents a major step forward. We show compelling evidence that an effective pushing force potential plays a role in aster interactions. With this critical knowledge, we can now explore for the potential molecular mechanisms – but such information lies beyond the current manuscript scope. This is particularly challenging due to the lack of specific microtubule drug inhibitors in *Drosophila*. We highlight related issues in the Discussion: paragraph starting on line 340 and lines 367-370.

Specific suggestions:Microtubules should be visualized more clearly (either in live or fixed samples). This is particularly important in Figure 4E and Video 4 (laser ablation experiment to create asymmetric asters).

Author Response: This is similar to Reviewer 1 final comment above. These experiments are very challenging and being able to see the microtubules with sufficient clarity is not straightforward. Given our controls and previous experience, we are confident we are ablating the microtubules.

Minor points:1. The authors explain the roles of microtubule asters in several model systems in the first paragraph of the introduction part. Please specify the species and/or cell types in each description.

Author Response: We have provided as suggested.

1. In lines 164 and 172, the citing figure numbers should be modified to Supplementary Fig. 1A and 1B, respectively.

Author Response: We thank the Reviewer for spotting this error. It has now been corrected.

1. The authors showed in the previous study that the boundary in the explant does not have an intact cell cortex and f-actin compartments (de-Carvalho et al., Development, 2022). This important informa8on should also be described in the current manuscript. It is also valuable to mention whether the pulling force mechanism operates in embryos where the intact cell cortex is present.

Author Response: This is an interesting point We have added a sentence in the discussion with this information. We have now added additional text in the Discussion (lines 324-327).